# A Literature Review on the Use of Artificial Intelligence for the Diagnosis of COVID-19 on CT and Chest X-ray

**DOI:** 10.3390/diagnostics12040869

**Published:** 2022-03-31

**Authors:** Ciara Mulrenan, Kawal Rhode, Barbara Malene Fischer

**Affiliations:** 1School of Biomedical Engineering & Imaging Sciences, Kings College London, London WC2R 2LS, UK; ciara.mulrenan@kcl.ac.uk (C.M.); kawal.rhode@kcl.ac.uk (K.R.); 2Rigshospitalet, Department of Clinical Physiology and Nuclear Medicine, Blegdamsvej 9, 2100 Copenhagen, Denmark; 3Department of Clinical Medicine, University of Copenhagen, 2100 Copenhagen, Denmark

**Keywords:** artificial intelligence, deep learning, medical imaging, SARS-CoV-2

## Abstract

A COVID-19 diagnosis is primarily determined by RT-PCR or rapid lateral-flow testing, although chest imaging has been shown to detect manifestations of the virus. This article reviews the role of imaging (CT and X-ray), in the diagnosis of COVID-19, focusing on the published studies that have applied artificial intelligence with the purpose of detecting COVID-19 or reaching a differential diagnosis between various respiratory infections. In this study, ArXiv, MedRxiv, PubMed, and Google Scholar were searched for studies using the criteria terms ‘deep learning’, ‘artificial intelligence’, ‘medical imaging’, ‘COVID-19’ and ‘SARS-CoV-2’. The identified studies were assessed using a modified version of the Transparent Reporting of a Multivariable Prediction Model for Individual Prognosis or Diagnosis (TRIPOD). Twenty studies fulfilled the inclusion criteria for this review. Out of those selected, 11 papers evaluated the use of artificial intelligence (AI) for chest X-ray and 12 for CT. The size of datasets ranged from 239 to 19,250 images, with sensitivities, specificities and AUCs ranging from 0.789–1.00, 0.843–1.00 and 0.850–1.00. While AI demonstrates excellent diagnostic potential, broader application of this method is hindered by the lack of relevant comparators in studies, sufficiently sized datasets, and independent testing.

## 1. Introduction

Coronaviruses are a group of RNA viruses that give rise to respiratory-tract and intestinal infections [1]. Gaining high pathogenic status during the severe acute respiratory syndrome (SARS-CoV) outbreak in 2002–2003, a new coronavirus emerged in Wuhan, Hubei province, China in December 2019 [2]. The virus was named ‘COVID-19’ or ‘SARS-CoV-2” and as a result of its rapid spread, was declared a pandemic by the World Health Organization (WHO) in March 2020 [3]. As of 3 January 2022, there have been a total of 291,721,552 cases worldwide, which is increasing at a steady rate each day [4]. The most commonly used diagnostic test is the nasopharyngeal swab for reverse-transcriptase polymerase chain reaction (RT-PCR). However, RT-PCR has lower than optimal sensitivity rates. At day 1, RT-PCR has a false-negative rate of 100%; by day 4 it lowers to a rate of 67% and reaches 38% by the time of symptom onset [5]. More recently, as mass testing has emerged, rapid lateral-flow tests have been used to detect COVID-19. The sensitivity of these tests is dependent on the skill of the individual performing the test: laboratory scientists perform with a sensitivity of 79%; in self-trained members of the public, this level is 58% [6].

It is pivotal that a diagnostic test demonstrates a high sensitivity rate, particularly for COVID-19, so that the infected individual is directed to self-isolate, thereby reducing transmission [7]. At present, the RT-PCR testing method is the only approved method to detect the COVID-19 disease [8]. It has been reported that medical imaging can be used to deliver a fast and accurate diagnosis in suspected COVID-19 patients. While it would not be practical to image everyone in the general population, it would be particularly useful for those seeking health-care services. This would allow for the rapid determination of the infection status of each patient, thereby directing their isolation protocol.

Both computed tomography (CT) and X-ray of the chest are routinely used for the diagnosis of a range of respiratory conditions, especially pneumonia. Chest X-rays are a quick diagnostic tool for respiratory conditions and are routinely used in emergency medicine, yielding a quick visualization of pathology. The portability of the imaging equipment makes this favorable, particularly in severe infections. CT is more sensitive in most settings, as well as reliable and fast, but more expensive and involves a higher radiation dose. CT has proven to be a useful tool for the imaging of COVID-19, allowing for the immediate detection of COVID-19. Characteristic imaging features of COVID-19 include peripheral ground-glass opacities (GGO) and multi-lobe consolidation [9]. Medical imaging, chest CT and X-ray have demonstrated significant success in detecting COVID-19 [10]. However, the COVID-19 pandemic has also highlighted how the health-care systems in many countries are under immense pressure and face many challenges, including a lack of trained radiologists. Thus, automated image analysis using artificial intelligence (AI) has been the focus of many studies published during this pandemic.

Prior to the pandemic, the use of AI in medical imaging had been rapidly evolving, i.e., in the past 10 years publications on AI in radiology have risen from around 100 per year to more than 700 per year. The ‘AI’ terminology is applied when a device mimics our own cognitive function, potentially minimizing the need for human intelligence or interaction in specific tasks. Within the field of medical imaging, deep learning has been rapidly and broadly applied using artificial neural networks to decode imaging data [11]. Learning can be unsupervised, whereby the algorithm will find patterns in unlabeled data, whereas supervised learning utilizes labels to aid in classification [12].

Deep learning (DL) is a subset of machine learning. It comprises multiple neural-network layers to extract higher-level features from the raw input. For instance, lower layers may identify edges and lines and higher layers identify more specialized features. The input into each layer successively utilizes the output of its preceding layer [13]. A specific subset of DL is explored in this review, i.e., convoluted neural networks (CNNs). A CNN can be trained to analyze information held within an image, and is thus built to process, segment, and consequently classify the image.

The purpose of this paper is to review and evaluate the published literature on the diagnostic performance of deep learning, i.e., CNNs, in medical imaging for COVID-19.

## 2. Materials and Methods

The following databases were searched: ArXiv, Google Scholar, MedRix and PubMed; all were searched up to the date of 5 April 2021. To extract the relevant articles on AI and COVID-19, the following search criteria were applied: ‘deep learning’, ‘artificial intelligence’, ‘medical imaging’, ‘COVID-19’ and ‘SARS-CoV-2’. As the publication tradition within the field of AI differs slightly from the traditional medical literature, the inclusion of non-peer reviewed articles from these databases was also allowed. Out of the articles found, only those that explored the use of AI with either CT or chest X-ray were selected for review. Out of these, it was assessed whether a DL algorithm was used.

From each of the papers the following data were extracted: the number of datasets used for training and validation, the proportion of COVID-19 scans within the dataset as well as the sensitivity, specificity, and AUC of the proposed method. It was also noted whether the datasets and model code were publicly available. The studies were then categorized by imaging modality: CT or chest X-ray. Only articles containing datasets with more than 100 images of confirmed cases of COVID-19 were included.

As an official framework with which to assess AI studies is yet to be published, each study in this review was assessed using a modified version of the Transparent Reporting of a Multivariable Prediction Model (TRIPOD) [14]. This reporting statement allows for the reporting of studies that develop, validate, or update a predictive model for diagnostic or prognostic purposes. The TRIPOD assesses the quality of the study in 6 areas (title and abstract, introduction, methods, results, discussion, other information). This includes adequate reporting of the study context, purpose (e.g., validation or development), source of data, information about participants, sample size, handling of missing data, and statistical analysis. Further, the adequate reporting of model development, performance and validation as well as limitations and study fundings are assessed.

The modified TRIPOD statement used in this review assesses 12 of 22 items that are most relevant to AI studies [15]. This modified statement applied the following: title, background and objectives, source of data, participants, outcome, sample size, participants, model performance, interpretation, implications, supplementary material, and funding. The outcome is summarized in Table 1 and Table 2.

In addition to the TRIPOD assessment, an additional clinical-relevance score was applied to all the included studies [16]:Clinical relevance uncertain.Potentially clinically relevant but not evaluated against a relevant comparator and lacks independent testing.Potentially clinically relevant and has demonstrated value against a relevant comparator but lacks independent testing.Potentially clinically relevant and has demonstrated value against a relevant comparator and has been independently tested.1–4 fulfilled and ready for implementation.

Only papers with a score of 2 or higher were included in this study.

**Table 1 diagnostics-12-00869-t001:** Modified TRIPOD assessment of X-ray studies.

Lead Author	Year	Country	Study Type	Aim	Dataset	Reference Standard	Comparator	Validation	External Testing	Main Findings	CRS	Funding
[17]	2020	United States of America	Case control	Detection of COVID-19 from X-ray	Training: 103 COVID-19 images (GitHub COVID image dataset), 500 non-COVID but pathological, 500 normal (Kaggle RSNA Pneumonia Detection Challenge dataset). Validation: 10 COVID-19, 10 pneumonia, 10 normal (Veteran’s administration)	X-ray pre-annotated by Radiographer	No	Cross-validation	Yes	Machine-learning algorithm can diagnose cases of COVID-19 from Chest X-ray.	2	Not disclosed
[7]	2020	United States of America	Case control	Detection of COVID-19 from X-ray	Training: 4698, Validation 523, Testing 580. 269 of the images COVID-19 (GitHub COVID image dataset), 3949 non-COVID pneumonia, 1583 normal.	X-ray pre-annotated by Radiographer	No	K-fold cross-validation	No	Machine-learning algorithm can diagnose cases of COVID-19 from chest X-ray.	2	No external funding
[18]	2020	Israel	Retrospective	Detection of COVID-19 from X-ray	Training: 2076 Testing: 350	X-ray annotated by Radiographer and positive RT-PCR test.	No	Cross-validation	No	Machine-learning algorithm can diagnose cases of COVID-19 from a portable chest X-ray.	2	Not disclosed
[19]	2020	United States of America	Case control	Detection of COVID-19 from X-ray	Training: 6324, Validation: 1574, Test 1970. 34% of images healthy, 28% non-COVID viral, 27% bacterial, 5% COVID-19, 4% TB.	X-ray pre-annotated by Radiographer	No	Cross-validation	No	Machine-learning algorithm can diagnose cases of COVID-19 from chest X-ray.	2	No external funding
[20]	2020	United States of America	Retrospective	Detection of COVID-19 from X-ray	COVID = 455 (Cohen, 2020) Normal = 532 Bacterial pneumonia = 492 Viral non-COVID pneumonia = 552 (Kaggle RSNA Pneumonia Detection Challenge dataset) Split 75 training 25 validation	X-ray annotated by Radiographer	No	Epoch K-fold cross-validation	No	Machine-learning algorithm can diagnose cases of COVID-19 from a portable chest X-ray.	2	No external funding
[10]	2020	Bangladesh	Case control	Detection of COVID-19 from X-ray	Training: 17,749, of which 232 COVID-19. Validation: 1501, of which 51 COVID-19 positive. Dataset (GitHub Dr Cohen, RSNA pneumonia detection Kaggle, COVIDx).	X-ray pre-annotated by Radiographer	No	K-fold cross-validation (10-fold).	No	Machine-learning algorithm can diagnose cases of COVID-19 from chest X-ray.	2	No external funding
[21]	2020	United States of America	Retrospective	Detection of COVID-19 from X-ray	Training and Testing split 75:25 randomly. In total: 455 = COVID-19 positive (GitHub Dr Cohen, Kaggle). 532 = Normal. 492 = Bacterial pneumonia. 552 = Non-COVID viral pneumonia.	X-ray pre-annotated by Radiographer and/or a positive RT-PCR test.	No	K-fold cross-validation (5-fold).	No	Machine-learning algorithm can diagnose cases of COVID-19 from a portable chest X-ray.	2	No external funding
[22]	2020	Brazil	Case control	Detection of COVID-19 from X-ray	Training: 5715 Validation: 653 6309 images in total. Non-COVID (Kaggle RSNA Pneumonia Detection Challenge dataset) COVID-19 (25 GitHub Dr Cohen), 180 Societa Italiana di Radiologia Medica, 248 Peshmerga Hospital Erbil.	Positive RT-PCR.	No	K-fold Cross-validation (10-fold)	No	Machine-learning algorithm can diagnose cases of COVID-19 from chest X-ray and a full clinical history/examination.	2	No external funding
[23]	2020	Turkey	Case control	Detection of COVID-19 from X-ray	Training: 80% Validation: 20% 125 COVID-19 (GitHub Dr Cohen), 500 normal, 500 pneumonias (chestX-ray8)	X-ray in recovered patients, pre-annotated by Radiographer.	No	K-fold Cross-validation (5-fold)	No	Machine-learning algorithm can diagnose cases of COVID-19 from chest X-ray.	2	Not disclosed
[8]	2020	Japan	Case control	Detection of COVID-19 from X-ray	Training: 410 COVID-19 (GitHub, Dr Cohen), 500 non-COVID (NIH, ChexPert) Validation: 62 COVID-19,	X-ray pre-annotated by Radiographer	No	K-fold Cross-validation (10-fold)	No	Machine-learning algorithm can diagnose cases of COVID-19 from chest X-ray.	2	No external funding
[24]	2020	United States of America	Case control	Detection of COVID-19 from X-ray	Training: 84 COVID-19 (Radiopaedia, Societa Italiana di Radiologia Medica, GitHub Dr Cohen), 83 Normal (Mooney et al.). Testing: 25 COVID-19, 25 Normal. Validation: 11 COVID-19, 11 Normal.	X-ray pre- annotated by Radiographer.	No	Cross-validation	No	Machine-learning algorithm can diagnose cases of COVID-19 from chest X-ray.	2	Not disclosed

**Table 2 diagnostics-12-00869-t002:** Modified TRIPOD assessment for CT studies.

Lead Author	Year	Country	Study Type	Aim	Dataset	Reference Standard	Comparator	Validation	External Testing	Main Findings	CRS	Funding
[25]	2020	United States of America	Case control	Detection of COVID-19 from chest CT, by discriminating between Ground Glass Opacities in COVID-19 and Milliary Tuberculosis.	606 COVID-19 303 Normal (datasets not named) Tested an external dataset: 112 images.	CT slides pre-annotated by Radiographer	No	Cross-validation	Yes	Machine-learning algorithm can diagnose cases of COVID-19 from chest CT.	2	Veteran Affairs Research Career Scientist Award, VA COVID Rapid Response Support, University of South Florida Strategic Investment Program Fund, Department of Health, Simons Foundation, Microsoft and Google.
[26]	2020	China	Case control	Detection of COVID-19 from chest CT.	Training: 1210 COVID-19, 1985 Non-COVID. Testing: 303 COVID-19, 495 Non-COVID. All images from CC-CCII (China Consortium of chest CT Image).	CT slides pre- annotated by Radiographer	No	Epoch (200) cross-validation	No	Machine-learning algorithm can diagnose cases of COVID-19 from chest CT.	2	Not disclosed
[27]	2020	India	Case control	Detection of COVID-19 from chest CT.	Training: 1984 Testing: 497 2482 CT scans, 1252 COVID-19 (Kaggle—Eduardo), 1230 for other pulmonary disease	CT slides pre-annotated by Radiographer	No	Epoch (30) Cross validation	No	Machine-learning algorithm can diagnose cases of COVID-19 from chest CT.	2	No external funding
[9]	2021 (newest version)	Iran	Case control	Detection of COVID-19 from chest CT.	Training: 3023 Non-COVID, 1773 COVID-19. Validation: 336 Non-COVID, 190 COVID-19. Testing: 744 Non-COVID, 3081 COVID-19.	CT slides pre-annotated by Radiographer, confirmed by 2 pulmonologists, correct clinical presentation, positive RT-PCR report.	Yes, 4 experienced radiologists.	Cross validation	Yes	Machine-learning algorithm can diagnose cases of COVID-19 from chest CT.	4	No external funding
[28]	2020	China	Case control	Detection of COVID-19 from chest CT.	Training: 499 COVID-19 Validation: 131 COVID-19	CT slides annotated by Radiographer	No	(100 epoch’s) cross validation.	No	Machine-learning algorithm can diagnose cases of COVID-19 from chest CT.	2	No external funding.
[29]	2020	United States of America	Case control	Detection of COVID-19 from chest CT.	Training: 657 COVID-19, 2628 Non-COVID.Validation: 120 COVID-19, 477 Non-COVID. Test: 266 COVID-19, 1064 Non-COVID. Mixed data obtained from GitHub and private Indian Hospital.	CT slides annotated based off RT-PCR reporting.	No	(50 epoch’s) cross validation	No	Machine-learning algorithm can diagnose cases of COVID-19 from chest CT.	2	Not disclosed
[30]	2020	China	Case control	Detection of COVID-19 from chest CT.	Training: 642 COVID-19, 674 Non-COVID. Validation: 124 COVID-19, 181 Non-COVID. Test: 154 COVID-19: 218 Non-COVID.	CT slides pre-annotated by 2 Radiographers (with 30+ years experience)	Yes, 8 radiologists (4 from COVID-19 hospitals, 4 from non-covid hospitals).	Cross Validation	Yes	Machine-learning algorithm can diagnose cases of COVID-19 from chest CT.	4	Not disclosed
[31]	2020	Belgium	Case Cotnrol	Detection of COVID-19 from chest CT.	Training: 80% Testing: 20% (CHU Sart-Tilman, CHU Notre Dame des Bruyeres).	CT slides annotated based off RT-PCR reporting	No (restrospectively assessed radiologists performance but not simultaneously with the algorithm).	K-Fold cross-validation (10-fold).	No	Machine-learning algorithm can diagnose cases of COVID-19 from chest CT.	2	Interreg V-A Euregio Meuse-Rhine, ERC grant, European Marie Curie Grant.
[32]	2020	South Korea	Case Control	Detection of COVID-10 from chest CT.	Training: 1194 COVID-19 (Wonkwang Hospital, Chonnam Hospital, Societa Italiana di Radiologia Medica), 2799 Non-COVID (inc. pneumonia, normal lung, lung cancer, non-pneumonia pathology—all from Wonkwang Hospital) External testing: 264 COVID-19(images of COVID-19 from recently published papers)	CT slides pre-annotated by Radiographer	No	K-Fold cross-validation (5-fold).	Yes	Machine-learning algorithm can diagnose cases of COVID-19 from chest CT.	2	National Research Foundation of Korea, Ministry of Science, ICT and Future Planning via Basic Science Research Program.
[33]	2020	United States of America	Case Control	Detection of COVID-19 from chest CT.	Training: 526 COVID-19, 533 Non-COVID Validation: 177 COVID-19, 151 Non-COVID. Testing: 326 COVID-19, 1011 N0n-COVID.COVID-19 data from single lefts: Hubei, 2 lefts in Milan, Tokyo, Syracuse. Non-COVID data: Lung image database consortium, Syracuse, National institute of health USA.	CT slides annotated based off RT-PCR reporting	No	Cross validation	Yes	Machine-learning algorithm can diagnose cases of COVID-19 from chest CT.	2	NIH Centre for Intervential Oncology, Intramural Research Program of the National Institutes of Health and the NIH Intramural Targeted Anti-COVID-19 Program.
[34]	2020	United States of America.	Case Control	Detection of COVID-19 from chest CT.	Training: 242 COVID-19, 292 Non-COVID. Tuning: 43 COVID-19, 49 Non-COVID. Testing: 134 COVID-19, 145 Non-COVID. All data obtained from 18 medical lefts in 12 provinces in China.	CT slides annotated based off 2 × RT-PCR reporting.	Yes, 2 radiologists.	Cross Validation	No	Demographical information (travel, exposure, patient age, sex, WBC count and symptoms) combined with output from machine-learning algorithm can diagnose cases of COVID-19.	3	US NIH grant.
[35]	2020	China	Retrospective	Detection of COVID-19 from chest CT.	Training: 1294 COVID-19, 1969 Non-COVID. Testing: 1235 COVID-19, 1964 Non-COVID. External testing: 2113 COVID-19, 2861 Non-COVID.	CT slides annotated by 5 radiographers.	Yes, 5 radiologists.	Cross validation	Yes	Not disclosed.	4	Not disclosed

## 3. Results

### 3.1. Study Selection

The electronic search resulted in 312 studies; when duplicates were removed, this number became 309. A total of 192 studies were excluded as irrelevant based on the title and abstract evaluation, and the remaining 117 papers were assessed in full for inclusion, from which a further 94 were excluded (See Figure 1: PRISMA flow chart detailing exclusion criteria). Once the evaluation was complete, 23 studies remained.

### 3.2. General Characteristics

Table 1 and Table 2 detail the general characteristics of all 23 studies included in this review, and Table 3 and Table 4 summarize the findings of the studies.

All of the studies were retrospective case-control or cohort studies. The majority were from the United States of America 10/23 (43%) and China 4/23 (17%). Other studies were from: Bangladesh (1/23), Belgium (1/23), Brazil (1/23), India (1/23), Iran (1/23), Israel (1/23), Japan (1/23), Korea (1/23) and Turkey (1/23). The majority of the studies received no external funding (9/23), or it was not disclosed (9/23); for those studies that did receive external funding, four were funded by national health bodies, and one was commercially funded.

Out of the 23 studies, 22 shared some of the same datasets (Table 5) as well as the same model structure (Table 6).

### 3.3. Aim and Methodology

All of the studies applied deep learning with image input for diagnosing COVID-19. The studies could be further divided according to the following objectives:The detection of and screening for COVID-19 (binary classification)Forming a differential diagnosis between COVID-19 pneumonia and other viral or bacterial pneumonia (multiclass classification).

Out of the 23 studies, 11/23 studies used a binary classification; of these, 2/23 characterized COVID-19/normal and 9/23 characterized COVID-19/non-COVID. The remaining 12 studies utilized multi-class classification, of which 5/23 characterized COVID-19/bacterial pneumonia/viral pneumonia. The last 7/23 characterized COVID-19/non-COVID infection/normal.

### 3.4. Reference Standard and Comparator

The reference standard for the studies was varied; 15/23 studies used a ground truth label based on a radiologist’s annotation, 2/23 used RT-PCR test results to assign labels, and 6/23 used a mix of both radiologist review and RT-PCR results.

Out of the 23 studies, four compared the performance of AI with a relevant comparator, i.e., a radiologist with varying years of experience.

### 3.5. Validation and Testing

Validation methods are used to assess the robustness of a proposed model. Internal validation utilizes data from the original training source and external validation tests the performance of the model on a dataset from a new independent source. All studies applied internal validation, where the dataset was split into two for training and testing. The majority of the studies operated with a train-and-test format, dividing the dataset in two. In addition, some studies performed k-fold cross validation. Out of the 23 studies, seven studies performed external validation using an independent dataset.

### 3.6. Clinical Relevance and Main Findings

All 23 studies were scored using the previously mentioned clinical-relevance score. The relevance score of 2 was the most common as most of the studies lacked a relevant clinical comparator and only compared the AI performance against other algorithms.

Out of the 23, 11 studies used chest X-rays to assess for COVID-19; all the chest-X-ray studies scored 2 in clinical relevance due to the lack of a relevant comparator, and 1/11 [17] performed external validation in an independent dataset.

The remaining 12/23 studies assessed the use of chest CT, of which 4/12 ([9]) included a relevant comparator. Therefore, four of the papers managed to score higher than 2 in terms of the clinical-relevance score. Out of these four papers, three papers ([30]) were allocated a score of 4 as they also included independent testing.

Studies using X-ray found overall good detection rates of GGO and lobular consolidation to deduce a diagnosis of COVID-19. The studies demonstrated diagnostic accuracy, with a sensitivity range of 79–100% and a specificity range of 92–99%. However, none of the X-ray studies included a relevant comparator, so it cannot be assessed whether the algorithm was on par with the diagnostic ability of a radiologist. Among the studies using CT, four included a relevant comparator [9]. The results of these studies are described below and in Table 7.

In the four studies that included a relevant comparator, the AI algorithm outperformed the radiologist. Only [34] reported one incidence where the radiologist outperformed the algorithm, i.e., the radiologist performed better at binary classification of pneumonia and non-pneumonia. The average experience of the radiologists in these four studies ranged from 6–11 years; the mean experience was 8.6 years. The use of an algorithm generally demonstrated an increase in both the sensitivity and specificity.

Two studies [9] supplied information about the time taken for the algorithm to assess a dataset/image versus that of the relevant comparator. Both studies reported much faster evaluation times with the algorithm performing up to 142 times quicker than a human.

When assessing model performance between the two imaging modalities, calculating the median and respective interquartile ranges, the performance of models using X-ray was more consistent (see Figure 2, a boxplot demonstrating the smaller interquartile range). This may be due to the ease of 2D image processing, and the lack of a requirement for segmentation layers in the network or slice selection. The reporting of significant differences between the performances of the model and the radiologist was only provided by [34], where there was no reported significant difference for the model used in the study. 

## 4. Discussion

The aim of this study was to provide an overview and evaluation of the literature published thus far on the utilization of AI on CT/chest-X-ray for the diagnosis of COVID-19.

The selected papers utilized deep-learning techniques with transfer learning, which assisted in making up for small-dataset limitations. Those studies that utilized transfer-learning methods were able to achieve high accuracies in diagnosis by employing previous image-analysis algorithms. All studies were assessed for potential bias, scored for clinical relevance, and evaluated using a modified TRIPOD assessment [15].

Compared to most AI studies in chest X-ray/CT, the datasets of those included in this review had a significantly smaller population; the average pathological dataset (in this case COVID-19 positive) for chest X-ray was: 340 images (range: 120–500, median: 360). Similarly, for chest CT the average dataset size was: 985 images (range 181–3084, median: 820). With fewer data available, the algorithms may not be trained or tested as thoroughly as in other AI studies. The average proportion of COVID-19 images in the datasets was 34%. This means that most of the dataset, 66%, is comprised of non/alternative-pathogenic images. This can potentially overestimate the sensitivity and positive predictive value of the algorithms as the proportion of positive cases in a typical clinical setting is likely to be much smaller.

All the included studies were retrospective studies. There is a lack of prospective testing, and no clinical trials were identified in the search. This is likely due to the recency of the COVID-19 pandemic rather than the newness of DL, as clinical studies are more prevalent in other applications of DL. One challenge that was common to all the studies was the small datasets available. As COVID-19 has only been around since the later part of 2019, there are relatively few images of COVID-19 patients available at individual institutions and public databases. A few of the studies even used the same databases (Table 6). This is a weakness, as the algorithms trained on a certain dataset may not be able to perform equally well when applied to different data [36]. For example, when data comes from only one demographic region, it may not perform as well on different demographics, which emphasizes the need for independent testing. This risk of bias is further enhanced by the lack of external validation among the papers reviewed.

Smaller datasets make it difficult to assess the reproducibility of the algorithm performance. While some studies pooled data from several public datasets, this does decrease transparency with regard to the origin and character of the image data. Another review of the application of AI to diagnose COVID-19 reported that the high risk of bias in most of their papers was a result of the small sample size of COVID-19 images [22]. However, small datasets are not exclusive to AI studies in COVID-19; an AI study on pulmonary nodules assessed an average sample size of images from only 186 patients [16].

Images included within the studies have been sourced from numerous public repositories and taken from publications [32]. It is likely that these images show extreme and interesting cases of COVID-19 that may be easier for the algorithm to detect. Further, in several studies datasets were expanded via image augmentation and the formation of iterations. Out of the 23 studies, only 6 performed independent testing with an external dataset (26%). Performance in externally validated studies was calculated at an average sensitivity and specificity of 93%. The average sensitivity and specificity in studies that did not externally validate their model were 92% and 94%. Thus, the externally validated models seem to work equally well on hitherto unseen data, which is reassuring. This lack of external validation has also been reported in another recent review on the topic [37]; only 13/62 (20%) assessed their algorithms on independent datasets. High performance in external testing proves that the model is generalizable to other patient populations, and in its absence it is difficult to tell how the model will perform if transitioned into clinical practice.

Yousefzadeh et al. performed external validation using data that the model had previously classified into multiple classes, but rediverted the images into binary groups [9]. There was one example where the model’s generalizability was tested, whereby a dataset of exclusively asymptotic patients was fed through the model. This selection of images was far more likely to be representative of those found in the community [25], permitting sound assessment of the model when reviewing images of less-extreme cases of COVID-19. The validation of a model on a set of low-quality images from recently published papers was also performed in order to assess the stability of the model [32], which could potentially be used for machines of poorer quality or for images captured with poor resolution/contrast.

Most studies lacked a relevant comparator. In this instance, the relevant comparator is not a PCR test, but a human comparator (i.e., radiologist) who is assessing the same image. Studies omitting a human comparator can cause the performance of the AI model to appear better than it is. Thus, it is important to contrast the performance of the new AI system to current practice prior to implementation in order to assess how the model may best serve in clinical practice. Only 5 of the 23 studies included a human comparator. With each study there was a varying degree of experience of the radiologist, which consequently influenced the degree of success the algorithm was perceived to have. Those with more experience rivaled the performance of the AI more closely, whereas junior radiologists may inflate the capability of the model. It is important for these studies to establish whether the machine is made to aid trainees, non-experts, or specialists. The average reader experience in the study by Mei et al. was just over 5 years senior to those in the paper by Liu et al., yet both yielded similar diagnostic accuracies. This suggests that the experience of the radiologist may not influence the ability to diagnose COVID-19, as it is a new pathology with new disease manifestations.

Studies will often aim for their algorithm to outperform that of a radiologist; however, it is important to note that an algorithm can still be of use even if it does not outperform an experienced radiologist. AI can still be used to lower the clinical burden, performing tasks with a similar accuracy at faster speeds. Studies by Yousefzadeh [9] and Jin [35] both assessed the speed at which their algorithms could perform, and both were much quicker than human analysis. In general, the included studies tended to pitch AI vs. human interpretation when perhaps a synergistic approach would have yielded greater benefit. Incorporating AI into a COVID-19 diagnosis could mean faster, more accurate diagnoses that incorporate various pieces of clinical information.

There are several limitations to this review. Following a comprehensive search for papers assessing the use of AI in reaching a COVID-19 diagnosis, it is possible that not all papers were included. As all publications on COVID-19 are new and further studies are published at a high speed, this review cannot claim to be up to date. In addition, a number of the papers at the time of writing are still pre-prints.

This review highlights a number of biases present in the literature, e.g., small sample size, potential for image duplication, differing image quality, extreme cases included in dataset, as also discussed by Roberts [37], and these biases limit the ability to translate AI into clinical practice.

As COVID-19 continues to pose a significant threat to health, more people are requiring both screening and testing. RT-PCR remains the current ‘gold standard’ for diagnosis; however, there are limitations in turnaround time, with test results taking anywhere between 3 h and 72 h, depending on price paid or priority assigned to turnaround. Rapid testing in the form of lateral-flow tests can bridge limitations in turnaround time and PCR supply, but they are unreliable for a diagnosis in primary care. A diagnosis in health care must be accurate in order to direct the isolation protocol and triage. AI programs have the potential to serve as an accurate and rapid aid in diagnosis.

AI can be developed to analyze the same findings that experienced radiologists can also extract. In addition, AI can also detect manifestations of disease that may not be obvious to the human eye, in turn increasing the sensitivity of image review. Once the limitations of small datasets, lack of relevant comparators and a clear standard reporting have been overcome [16], the use of AI can be extended beyond just formulating a diagnosis, it can also make predictions about the course and severity of the disease. Some of the AI models can match similar presentations with those it has previously assessed and share information about the experience of the disease course and outcome. It is essential that sensitivity rates of these AI models are high in all incidences in order to ensure there are no false negatives, and that everyone needing to self-isolate is informed to do so. AI is also able to monitor the long-term manifestations of lung diseases. If AI can be implemented alongside ‘traditional’ methods of diagnosis, then perhaps faster, definitive, and accurate instructions can be determined for self-isolation protocol and identifying patients at high risk.

## 5. Conclusions

This review summarizes the published research on AI systems for COVID-19 detection on CT and chest X-ray. The presented studies report promising results for the automated diagnosis of COVID-19 by both modalities using deep-learning methods. However, while AI shows a promising diagnostic potential, this area of research does suffer from small datasets as well as the lack of a relevant clinical comparator and external validation, giving rise to a high risk of bias that limits its transferability into clinical practice. Thus, future research should include relevant clinical comparison and external validation in order to increase the likelihood of new AI systems being deployed in fields that are of the greatest benefit to patients.

## Figures and Tables

**Figure 1 diagnostics-12-00869-f001:**
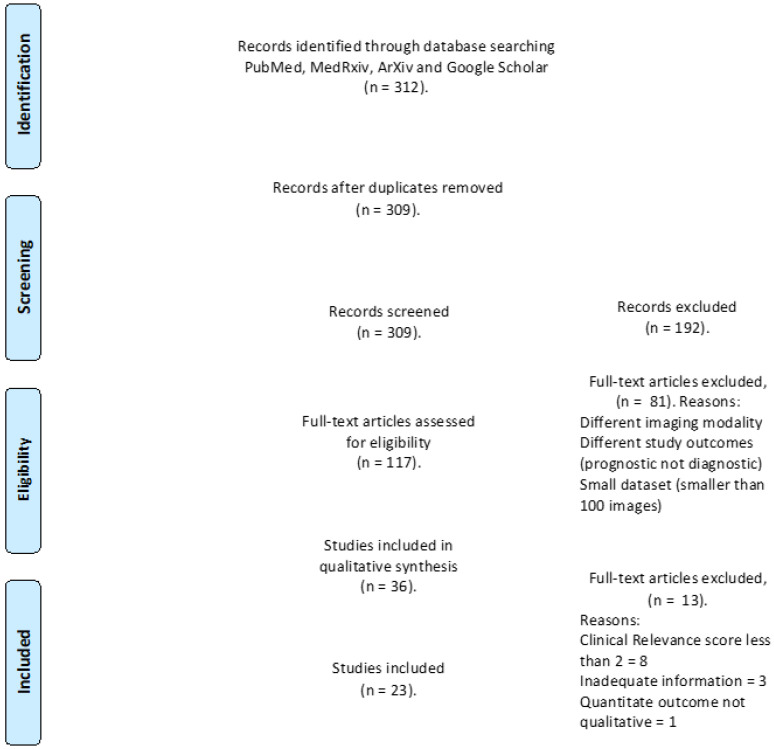
PRISMA flow chart displaying the selection process of chosen studies, including total identified records, number screened, number of duplicates, included and excluded records for inclusion.

**Figure 2 diagnostics-12-00869-f002:**
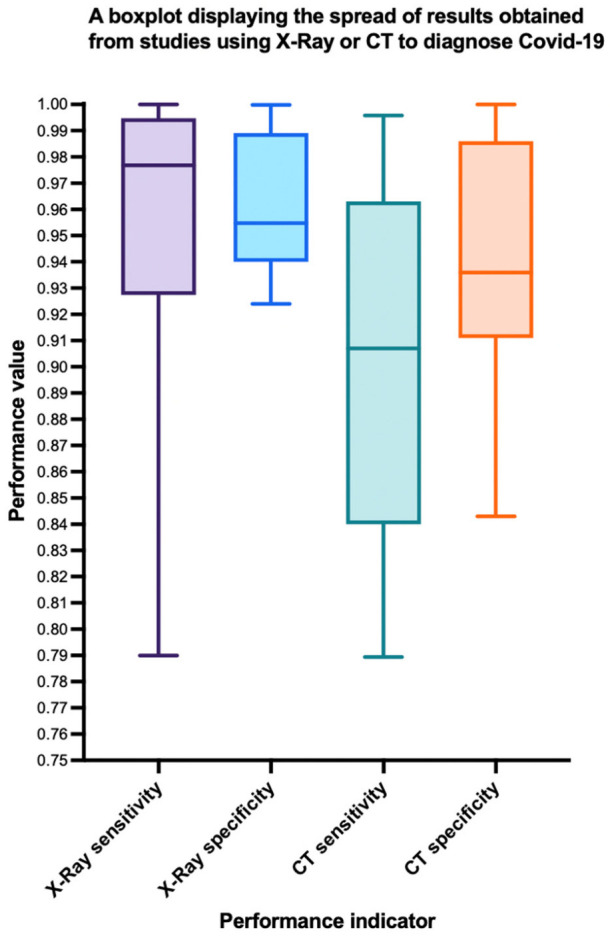
Box plot displaying the spread of results from CT and X-ray models. Interquartile range is demonstrated by the box, with the median value in the center. Whiskers display extremes at either ends.

**Table 3 diagnostics-12-00869-t003:** Summary of results from AI Studies for chest X-ray COVID-19 classification.

Author	Dataset	Deep Learning Model	2D/3D	All Data	COVID Data	Train All/COVID	Validation All/COVID	TestAll/COVID	Sensitivity	Specificity	AUC	Dataset	Code URL
[17]	COVID-19/non-COVID pneumonia/normal	Microsoft CustomVision	2d	1000	500	970/474	30/10	/	100	95	/	Public	*/*
[7]	COVID-19/non-COVID infection/normal	AIDCOV using VGG-16	2d	5801	269	4698	523	580	99.3	99.98	n/a	Public	*/*
[18]	COVID-19/normal	RetNet50	2d	2427	360	2120/84	350/178	n/a	87.1	92.4	0.94	Institutional dataset	*/*
[20]	COVID-19/viral pneumonia/normal	VGG-16	2d	1034	274	724/192	/	310/82	/	/	0.9978	Public	*/*
[19]	COVID-19/viral/bacterial pneumonia/TB	DenseNet-201	2d	9868	494	6234/(n/a)	1574/(n/a)	1970/125	94.62	/	/	Public	*/*
[8]	COVID/non-COVID	HRNet	2d	1410	410	n/a	/	1410/410	98.53	98.52		Public	*/*
[21]	COVID/bacterial pneumonia/viral pneumonia/normal	VGG-16	2d	2031	445	1523/334/	/	508/111	79.0	93	0.85	Public	*/*
[10]	COVID/non-COVID	Faster R-CNN	2d	19,250	283	17,749/232	/	1501/51	97.65	95.48	/	Public	/
[22]	COVID-19/bacterial and viral pneumonia/normal	IKONOS	2d	6320	464		/	/	97.7	99.3	/	Public	https://github.com/Biomedical-Computing-UFPE/Ikonos-X-Desktop
[32]	COVID-19/pneumonia/normal	DarkNet-19	2d	1125	125	900/100	/	225/25	95.13	95.3	/	Public	https://github.com/muhammedtalo/COVID-19
[24]	COVID-19/non COVID-19	Residual Att Net	2d	239	120	167/84	50/25	22/11	100	96	1	Public	https://github.com/vishalshar/COVID-19-screening-using-RAN-on-Xray-images

**Table 4 diagnostics-12-00869-t004:** Summary of results from AI Studies for CT COVID-19 classification.

Author	Dataset	Deep Learning Model	2D /3D	All Data	COVID Data	Train All/COVID	Validation All/COVID	Test All/COVID	Sensitivity	Specificity	AUC	Dataset	Code URL
[25]	COVID-19/non-COVID disease/normal		2d	904	606	2685/2116	/	34/34	97.06	/	0.9664	Institutional dataset	/
[26]	COVID-19/common pneumonia/normal	MNas3DNet41	3d	3993	1515	3195/1213	/	798/302	86.09	93.15	0.957	Public	/
[27]	COVID-19/non-COVID-19 disease	GLCM	2d	2483	1252	1986/1002	/	497/250	/	98.77	0.987	Public	/
[9]	COVID-19/non-COVID pathological/normal	Ai-corona	2d	2121	720	1764/601	/	357/119	92.4	98.3	0.989	Institutional dataset	https://ai-corona.com/
[28]	COVID-19/normal	DeCoVNet	3d	630	630	499	/	131	90.7	91.1	0.976	Institutional dataset	https://github.com/sydney0zq/COVID-19-detection
[30]	COVID/non-COVID	COVIDNet	2d	1993	920	1316/642	522/233	894/387	92.2	98.6	0.98	Institutional dataset	/
[31]	COVID/non-COVID	RadiomiX	2d	1381	181	1104/145	/	276/36	78.94	91.09	0.9398	Institutional dataset	/
[32]	COVID-19/bacterial/viral pneumonia	FCONet ft ResNet-50	2d	4257	1194	3194/955	/	1063/239	99.58	100	1	Institutional dataset e	/
[33]	COVID-19/non COVID-19	Densenet-121	3d	2724	1029	1059/526	328/177	1337/326	84.0	93.0	0.949	Mixed	https://ngc.nvidia.com/catalog/containers/nvidia:clara:ai-COVID-19
[34]	COVID-19/non-COVID-19	Inception-ResNet-v2	3d	905	419	534/242	92/43	279/134	82.8	84.3	0.92	Institutional dataset	https://github.com/howchihlee/COVID19_CT
[35]	COVID-19/viral pneumonia/bacterial pneumonia/influenza	OpenCovidDetector	2d	11,356	3084	2688/751	/	6337/2333	87.03	96.60	0.9781	Public	https://github.com/ChenWWWeixiang/diagnosis_covid19
[29]	COVID-19/non COVID-19	U-Net	2d	5212 (slices)	275	3285/657	597/120	1330/266	96.3	93.6	/	Public	/

**Table 5 diagnostics-12-00869-t005:** The share of datasets amongst studies.

Dataset Name	Study Used
Kaggle RSNA Pneumonia Detection challenge dataset	[8,17]
NIH	[33]
SUNY	[33]
LIDC	[33,35]
CC-CCII	[35]
Tianchi-Alibaba	[35]
MosMedData	[35]
Cohen database	[10,17,21]
Italian society of Medical and Interventional Radiology	[32]
WKUH (Wonkwang University hospital)	[32]
CNUH (Chonnam National University Hospital)	[32]
COVID-19-CT-dataset	[29]
MDH (MasihDaneshvari Hospital)	[9]
Peshmerga Hospital Erbil	[22]

**Table 6 diagnostics-12-00869-t006:** Deep learning methods (CNN) used across all studies.

Network Name	Study Used
HRNetHRNet	[8]
Microsoft CustomVisionMicrosoft CustomVision	[17]
GLCMGLCM	[27]
ResNetResNet	[18,24,32,34,35]
RadioMixRadioMix	[31]
UNetUNet	[28,29]
VGGVGG	[7,10,20,21,24]
DenseNetDenseNet	[19,26,30,33]
DarkNetDarkNet	[23,25]
EfficientNetEfficientNet	[9]

**Table 7 diagnostics-12-00869-t007:** Overview of AI and radiologist performance in studies with radiologist as comparator.

Study	AI Performance		Radiologist Performance		Additional Information	Experience of the Radiologist
	Sensitivity	Specificity	Sensitivity	Specificity		
[9]	0.92	0.97	0.90	0.88	Time taken to assess one image: AI→2.02 s. Radiologist→58.0 s.	4 radiologists, average experience 9.25 years.
[30]	0.92	0.99	0.77	0.90		4 radiologists, average experience 11.25 years.
[35]	0.98	0.91	0.96	0.72	Time taken to assess one image: AI→2.73 s. Radiologist→390 s. All 5 radiologists had a n average of 8 years experience.	5 radiologists, average experience of 8 years.
[34]	0.84	0.83	0.75	0.94		2 radiologists, average experience of 6 years.

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
