# Peer review of "A Literature Review on the Use of Artificial Intelligence for the Diagnosis of COVID-19 on CT and Chest X-ray"

_diagnostics, 2022, doi:10.3390/diagnostics12040869_

Round 1

Reviewer 1 Report

Dear authors, thank you for submitting this very interesting review about AI in detecting covid19 pneumonia through chest x-ray and/or CT scan. Although the limitations you have explained, I think this work could be really interesting for publication in this journal.

Author Response

We thank the reviewer for the kind comment.

Reviewer 2 Report

Need elaborate more in conclusion section

Author Response

Thank you for your useful comment. While we believe a conclusion should be relatively short and to the point, we have elaborated on the implications for future research. We have also expanded the Discussion-section.

Reviewer 3 Report

Introduction
  1. Provide more information on the medical imaging modalities of interest (i.e. CT and X-ray), It can be a paragraph each. One detailing CT and the other detailing X-ray.
  2. The principle of operation of both CT and X-ray should be added to each paragraph in (1).
  3. Provide your reason(s) for considering only papers that utilized only CT and X-ray. Provide the benefit and side effects of each imaging modality.
  4. All abbreviations must be first spelled out before any subsequent of them. AI should be spelled out in full before any subsequent use of it.
  5. There was no mention of Convolutional Neural Network (CNN) in the introduction section. The deep learning technique implemented in all of the studies is the CNN. So, include a paragraph in the introduction section on CNN.
Material and Methods
  1. Provide more information on the TRIPOD system.
  2. Are the paper used peer-reviewed or not? 
  3. Information about whether data pre-processing techniques implemented for the study reviewed should be provided.
Result
  1. Tables and Figures should be present in the section they are mentioned in. Why are Tables 1 to 4 not in the section they are mentioned in?

Author Response

We thank the reviewers for their generous comments on the manuscript and have edited the manuscript to address their concerns. Please see our point-by-point response in italics below:

Introduction

  1. Provide more information on the medical imaging modalities of interest (i.e. CT and X-ray), It can be a paragraph each. One detailing CT and the other detailing X-ray.
  2. The principle of operation of both CT and X-ray should be added to each paragraph in (1).
  3. Provide your reason(s) for considering only papers that utilized only CT and X-ray. Provide the benefit and side effects of each imaging modality.

Information on CT and X-ray and its use is now included in the introduction. Further, explanations for the exclusive inclusion of CT and X-ray has been included. Primary focus on CT and X-ray was decided by their routine application in the diagnosis for the range of conditions the AI was developed to distinguish between (for instance: pneumonia and influenza). Additional information on the benefits and limitations of the modalities have briefly been considered and noted in the text as suggested.

  1. All abbreviations must be first spelled out before any subsequent of them. AI should be spelled out in full before any subsequent use of it.

Sorry about this, all abbreviation of terms has been noted and corrected accordingly.

  1. There was no mention of Convolutional Neural Network (CNN) in the introduction section. The deep learning technique implemented in all of the studies is the CNN. So, include a paragraph in the introduction section on CNN.

We completely agree and this has now been included in the introduction.

Material and Methods

  1. Provide more information on the TRIPOD system.
  2. Are the paper used peer-reviewed or not? 
  3. Information about whether data pre-processing techniques implemented for the study reviewed should be provided.

A more comprehensive explanation of the TRIPOD reporting statement has been included, and each reporting division for the model can be found in Table 1 and 2 as presented in the results section. We have clarified the uncertainty with regard to inclusion of non-peer reviewed articles in the methods section. With regard to details on data-preprocessing techniques we have not included this because we wanted to keep a clear clinical focus and allowing clinicians without detailed knowledge on the world of AI to get an impression of the status in this field.

Result

  1. Tables and Figures should be present in the section they are mentioned in. Why are Tables 1 to 4 not in the section they are mentioned in?

As suggested, the tables and figures have been integrated into the manuscript chronologically.

Round 2

Reviewer 3 Report

The information provided on the TRIPOD system is not sufficient. More detailed information is needed.

Author Response

We have now included more details on the TRIPOD assessment (p3).

Round 3

Reviewer 3 Report

Good job. All highlighted corrections have been attended to and resolved.